# EVALUATING VISUAL "COMMON SENSE" USING FINE-GRAINED CLASSIFICATION AND CAPTIONING TASKS

**Raghav Goyal, Farzaneh Mahdisoltani, Guillaume Berger, Waseem Gharbieh, Ingo Bax, Roland Memisevic**
Twenty Billion Neurons Inc.
`{firstname.lastname}@twentybn.com`

## 1 INTRODUCTION

Understanding concepts in the world remains one of the well-sought endeavours of ML. Whereas ImageNet enabled success in object recognition and various related tasks via transfer learning, the ability to understand physical concepts prevalent in the world still remains an unattained, yet desirable, goal. Video as a vision modality encodes how objects change across time with respect to pose, position, distance of observer, etc.; and has therefore been researched extensively as a data domain and for studying "common sense" physical concepts of objects.

The current standard approach to video data collection uses videos from existing resources on the web such as YouTube, Vimeo or Flickr that are passed through a pipeline of annotators to gather labels. Instances of this approach include the Kinetics (Kay et al., 2017) or Moments in Time (Monfort et al.) datasets. Since this approach provides for limited control of variations in pose, motion and other aspects of the objects relevant to learning, recently novel datasets based on "*crowd-acting*" have emerged, where crowd workers are asked to generate videos according to pre-defined labels. Examples include the first version of the something-something (Goyal et al., 2017) dataset and the Charades dataset (Sigurdsson et al., 2017).

In this work, we describe an updated version of the something-something dataset and a variety of novel experiments on this data. The first version of this dataset, that was introduced by (Goyal et al., 2017), is generated by asking crowd workers to act out template-based labels, such as "Dropping *something* into *something*" and to fill in the "something" placeholders with the object used to generate the video. The work is aimed specifically at learning physical aspects of the world rather than focusing on action detection or recognition. In this paper, we build on and extend that work through the following contributions: First, we release version 2 of the dataset - *something-something-v2*[1], containing $220,847$ videos across $174$ action classes and in contrast to the first version of the dataset containing *object categories*. Second, we propose an approach to quantify progress on the dataset constructed using contrastive examples. Third, we present baseline results, which show surprisingly strong performance on the arguably hard discrimination tasks represented by this metric. Fourth, we show how temporally-sensitive saliency maps can be used to visualize regions of interest within a video and reveal the dependency of subtle classification decisions on spatio-temporal patterns in the video. Fifth, we show captioning results on the data, as predicting the expanded labels containing object categories can be viewed as a (very hard) captioning task.

## 2 PROPOSED COMMON SENSE METRIC

We argue that for datasets which focus on fine-grained visual aspects, a metric is needed that can evaluate discriminating properties of the model apart from top-k accuracy. As a first step, for something-something-v2, we looked at the topmost confusions from a baseline model (described later in Section 3), and found that the confusions are ambiguous enough even for humans to differentiate, and these are inevitably accounted in the accuracy. For e.g. $55.42\%$ of the samples from "*Moving something across a surface until it falls down*" label are confused with "*Pushing something so that it falls off the table*" (Please refer to Table 3 in Appendix for a list of top 10 confusions).

---

[1]We plan to make the dataset available after acceptance of the paper

**Contrastive groups**    In order to avoid models picking up indirect cues from object type, hand position, camera shake, etc., we propose to measure performance by forming groups of classes, which are superficially similar (but distinct) and easily confused by networks. Many of the classes within a group contain multiple similar actions with minute visual differences such that fine-grained understanding and attention to detail is necessary to yield correct classification. For example, the actions "Throwing something in the air and catching it" and "Throwing something in the air and letting it fall" belong to one contrastive group. The following aspects went into the definition of the action groups:

***Pretending classes***: Action classes are grouped with their corresponding "pretending" classes if it exists ***Aggregating similar classes***: To reduce ambiguity, we combine semantically similar classes into a single class, e.g. "Moving something across a surface until it falls down" and "Pushing something so that it falls off the table" ***Antonyms***: e.g. Digging/Burying, Folding/Unfolding, Dropping/Putting, Collide/Pass, Opening/Closing. ***Different object properties***: e.g. "Falling like a rock/paper", "Wringing something wet/twisting something". ***Different prepositions***: Classes differentiating between relative position of an object, e.g. behind/in front of/next to, into/onto, etc. ***Different final state***: Classes differentiating between final state of an object, e.g. tearing a bit/into two pieces, poking so that a stack collapses/doesn't collapse, continues spinning/stops spinning, catching/letting it fall, falls/doesn't fall down, etc.

In total we formed 69 contrastive groups, a full list can be found in Appendix in Table 2. We excluded classes containing concepts that we deem are fairly easy to distinguish, such as ***Relative motions***: e.g. left/right, camera movements (up/down, approaching/moving away), etc. ***Static***: Classes that have no motion, e.g. Showing, Holding, Squeezing, etc.

**Metric construction**    For each group, we filter examples from the validation set corresponding to only the classes contained in the group and compute the accuracy. We average the accuracy across all the groups to obtain the mean accuracy (mA1). To account for class imbalance we also compute the accuracy that follows from always predicting the most common class in the group (mA2).

To arrive at the final metric, we normalize the above scores, computed as a percentage difference between mA1 and mA2: $(mA1 - mA2)/(100 - mA2)$. The metric measures how well the model performs compared to predicting the most common class within each group. Table 1 shows the scores on the validation set. We refer to the normalized score on the contrastive groups as *Common sense score*. The Table also shows performance on the action groups (which are not designed to measure performance on detailed prediction tasks) proposed in (Goyal et al., 2017).

## 3    EXPERIMENTS AND RESULTS

### 3.1    PERFORMANCE OF MODELS

We compare a baseline model on something-something-v2 with the features from models pre-trained on well-established datasets - ImageNet and Kinetics. We use a length of 72 frames per video, covering a max duration of 6 secs at 12 fps, and padding short videos with their last frame.

**Baseline Model**    We use a VGG-style 3D-CNN with 11 layers of 3D convs, operating on $84 \times 84$ frames. It is trained from scratch on something-something-v2 and is referred to as *smth_3D_scratch*.

**Pre-trained models**    We took a Kinetics pre-trained I3D model[2] trained on RGB stream (Carreira & Zisserman, 2017) and extracted "Mixed_5c" layer features from the something-something-v2 dataset; and trained a 3D convolution and a fully-connected layer on top for classification. The model is referred to as *kinetics_feat_smth*.

We took an ImageNet-trained ResNet-152 model and extracted features for each frame in something-something-v2 dataset, and trained a 1D convolution and a fully-connected layer on top for classification, referred to as *imagenet_feat_smth*. Surprisingly, we find that despite the extraordinarily hard tasks inherent in this grouping, even the comparably simple baseline model yields performance that is significantly beyond chance. In the next section, we investigate the dependence of the classification decisions on local aspects of the input video.

---

[2]https://github.com/deepmind/kinetics-i3d

| Method | Accuracy (%) | | Common sense score (%) | |
|---|---|---|---|---|
| | top-1 | top-5 | Actual mA1 (mA2) | Normalized |
| imagenet_feat_smth | 22.67 | 48.32 | 74.81 (64.93) | 28.17 |
| kinetics_feat_smth | 27.93 | 55.75 | 77.13 (64.93) | 34.78 |
| smth_3D_scratch | 50.28 | 79.51 | 88.90 (64.93) | 68.34 |

Table 1: Comparison among models trained from scratch and pre-trained features (from Kinetics and ImageNet) fine-tuned on something-something-v2 dataset.

## 3.2 SALIENCY MAPS

To visualize the regularities that models learned from the data, we extracted *temporally-sensitive* saliency maps using Grad-CAM (Selvaraju et al., 2017). We extended the implementation[3] in time dimension and projected back the weighted activation maps to the input space. Figure 1 shows saliency maps of examples predicted as "*Opening something*". Some additional visualizations of the feature space learned by the model using T-SNE are shown in the appendix.

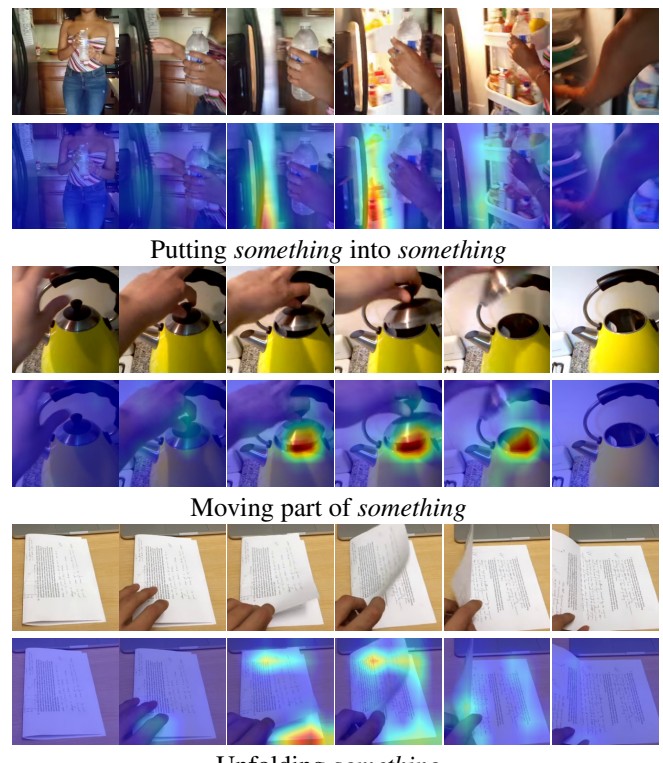

Putting *something* into *something*

Moving part of *something*

Unfolding *something*

Figure 1: Video examples predicted as "*Opening something*" having different ground-truth labels.

## 3.3 VIDEO CAPTIONING

Along with each video in something-something-v2, a string caption is provided which describes the video in a natural language sentence. As baseline, we use a basic encoder-decoder model for captioning something-something videos, which models the conditional probability distribution over word sequence $c$ given the video $v$ $p(c|v) = \sum_{i=1}^{m} \log p(c_{i+1}|c_{\leq i}, h)$. The encoder is a 3D-convnet similar to the one described above. Please refer to the appendix for some captioning examples.

---

[3] https://github.com/jacobgil/pytorch-grad-cam

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

# A   CONTRASTIVE GROUPS LIST

Table 2: List of 69 contrastive groups. Please note that some of the classes are aggregated together using "+" to represent one class, since they convey semantically similar concepts (e.g. group 25).

| # | Contrastive groups |
|---|---|
| 1 | Wiping something off of something
Pretending or failing to wipe something off of something |
| 2 | Closing something
Pretending to close something without actually closing it |
| 3 | Opening something
Pretending to open something without actually opening it |
| 4 | Picking something up
Pretending to pick something up |
| 5 | Turning something upside down
Pretending to turn something upside down |
| 6 | Putting something into something
Pretending to put something into something |
| 7 | Putting something behind something
Pretending to put something behind something |
| 8 | Putting something next to something
Pretending to put something next to something |
| 9 | Putting something on a surface
Pretending to put something on a surface |
| 10 | Putting something onto something
Pretending to put something onto something |
| 11 | Putting something underneath something
Pretending to put something underneath something |
| 12 | Scooping something up with something
Pretending to scoop something up with something |
| 13 | Throwing something
Pretending to throw something |
| 14 | Taking something out of something
Pretending to take something out of something |
| 15 | Spreading something onto something
Pretending to spread air onto something |
| 16 | Sprinkling something onto something
Pretending to sprinkle air onto something |
| 17 | Pouring something out of something
Pretending to pour something out of something, but something is empty |
| 18 | Taking something from somewhere
Pretending to take something from somewhere |
| 19 | Twisting (wringing) something wet until water comes out
Twisting something |
| 20 | Tearing something into two pieces
Tearing something just a little bit |
| 21 | Opening something
Closing something |
| 22 | Poking a stack of something so the stack collapses
Poking a stack of something without the stack collapsing |
| 23 | Attaching something to something
Trying but failing to attach something to something because it doesn't stick |
| 24 | Folding something
Unfolding something |

| 25 | (Moving something across a surface until it falls down + Pushing something so that it falls off the table) (Moving something across a surface without it falling down + Pushing something so that it almost falls off but doesn't) |
|----|---|
| 26 | Covering something with something Uncovering something |
| 27 | Throwing something in the air and catching it Throwing something in the air and letting it fall |
| 28 | Lifting up one end of something without letting it drop down Lifting up one end of something, then letting it drop down |
| 29 | Lifting something up completely without letting it drop down Lifting something up completely, then letting it drop down |
| 30 | Lifting a surface with something on it but not enough for it to slide down Lifting a surface with something on it until it starts sliding down |
| 31 | Pulling two ends of something so that it gets stretched Pulling two ends of something so that it separates into two pieces |
| 32 | Moving something and something so they collide with each other Moving something and something so they pass each other |
| 33 | Something falling like a feather or paper Something falling like a rock |
| 34 | Spinning something so it continues spinning Spinning something that quickly stops spinning |
| 35 | Putting something that can't roll onto a slanted surface, so it slides down Putting something that can't roll onto a slanted surface, so it stays where it is |
| 36 | Something colliding with something and both are being deflected Something colliding with something and both come to a halt |
| 37 | Plugging something into something Plugging something into something but pulling it right out as you remove your hand |
| 38 | Tilting something with something on it slightly so it doesn't fall down Tilting something with something on it until it falls off |
| 39 | Burying something in something Digging something out of something |
| 40 | Poking a hole into some substance Poking a hole into something soft |
| 41 | (Poking something so it slightly moves + Poking something so lightly that it doesn't or almost doesn't move) Poking something so that it falls over Poking something so that it spins around |
| 42 | Pouring something into something Trying to pour something into something, but missing so it spills next to it |
| 43 | Pouring something into something Pouring something into something until it overflows Pouring something onto something |
| 44 | Burying something in something Covering something with something |
| 45 | Covering something with something Digging something out of something |
| 46 | Bending something so that it deforms Bending something until it breaks |
| 47 | Dropping something behind something Dropping something in front of something |
| 48 | Dropping something into something Dropping something onto something |
| 49 | Dropping something behind something Putting something behind something |
| 50 | Dropping something in front of something Putting something in front of something |

| 51 | Dropping something into something
Putting something into something |
|---|---|
| 52 | Dropping something next to something
Putting something next to something |
| 53 | Dropping something onto something
Putting something onto something |
| 54 | Picking something up
Putting something on a surface |
| 55 | Failing to put something into something because something does not fit
Putting something into something |
| 56 | Scooping something up with something
Picking something up |
| 57 | Tipping something over
Touching (without moving) part of something |
| 58 | Tipping something over
Tipping something with something in it over, so something in it falls out |
| 59 | Tipping something with something in it over, so something in it falls out
Touching (without moving) part of something |
| 60 | Putting something behind something
Putting something in front of something
Putting something underneath something
Putting something next to something
Putting something onto something
Putting something into something |
| 61 | Putting something similar to other things that are already on the table
Taking one of many similar things on the table |
| 62 | Putting something into something
Taking something out of something |
| 63 | Putting something on a surface
Putting something on the edge of something so it is not supported and falls down |
| 64 | Laying something on the table on its side, not upright
Putting something that cannot actually stand upright upright on the table, so it falls on its side
Putting something upright on the table |
| 65 | Pretending to put something behind something
Pretending to put something next to something
Pretending to put something underneath something
Pretending to put something into something
Pretending to put something onto something |
| 66 | Putting something on a flat surface without letting it roll
Rolling something on a flat surface |
| 67 | Spilling something behind something
Spilling something next to something
Spilling something onto something |
| 68 | (Letting something roll along a flat surface +
Rolling something on a flat surface)
Putting something on a flat surface without letting it roll |
| 69 | Putting something onto a slanted surface but it doesn't glide down
Letting something roll down a slanted surface |

# B    TOP-10 CONFUSIONS

Table 3: Top 10 confusions from the baseline model

| True class | Predicted class | % of true instances predicted |
|---|---|---|
| Moving something across a surface until it falls down | Pushing something so that it falls off the table | 55.42 |
| Letting something roll along a flat surface | Rolling something on a flat surface | 44.77 |
| Pushing something off of something | Pushing something so that it falls off the table | 41.67 |
| Throwing something onto a surface | Throwing something | 39.02 |
| Poking a stack of something without the stack collapsing | Poking something so lightly that it doesn't or almost doesn't move | 37.50 |
| Lifting a surface with something on it but not enough for it to slide down | Tilting something with something on it slightly so it doesn't fall down | 36.84 |
| Spilling something next to something | Pouring something into something | 36.67 |
| Pouring something into something until it overflows | Pouring something into something | 36.00 |
| Trying to pour something into something, but missing so it spills next to it | Pouring something into something | 32.50 |
| Pretending to take something from somewhere | Pretending to pick something up | 31.20 |

# C    T-SNE PLOTS

Using Tensorboard's[4] implementation of t-SNE, we ran the optimization on validation set features (24777 examples, each with 512 dimensions) for 4k iteration with perplexity of 50 and learning rate of 10. A 2D representation is shown in the Figure 2.

---

[4]https://github.com/tensorflow/tensorboard

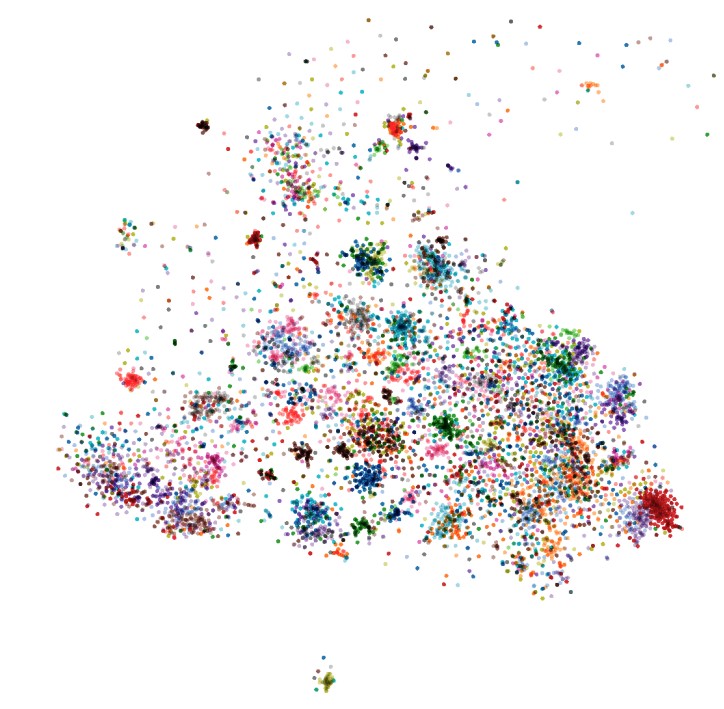

Figure 2: t-SNE of learned representations

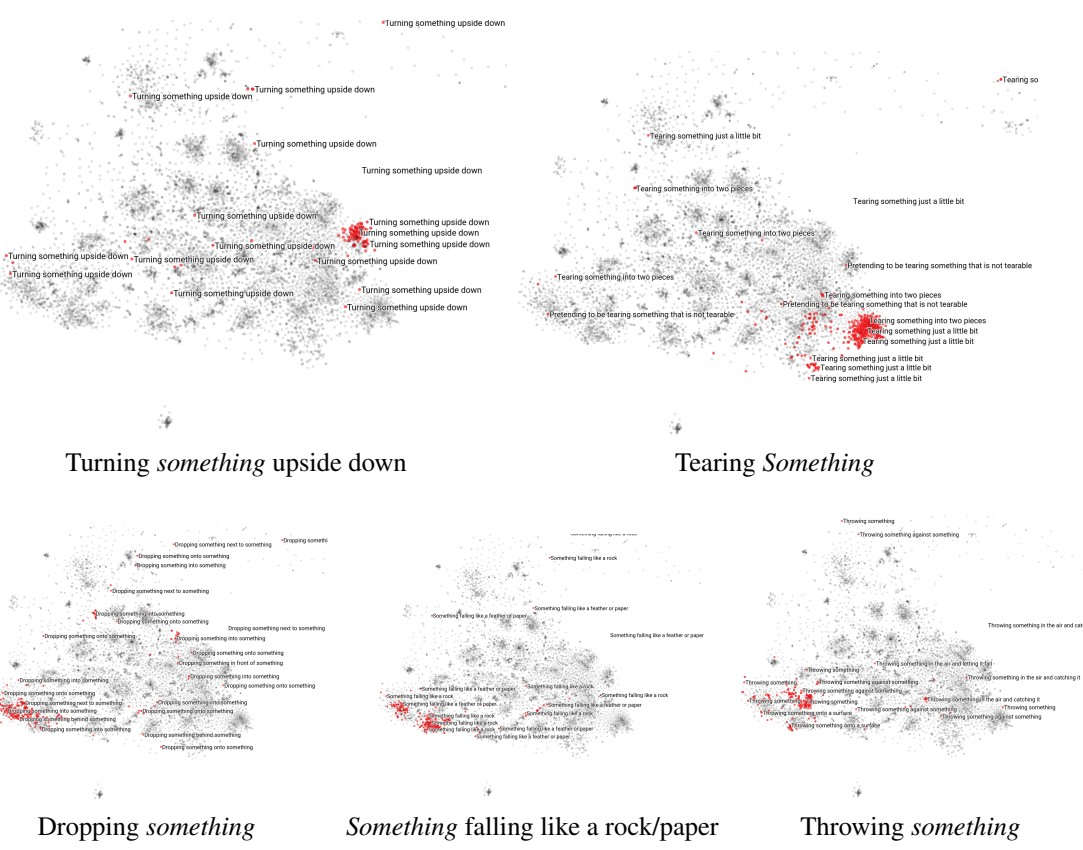

Figure 3: The above three classes are mapped to regions "near" to each other and also convey semantically similar meaning

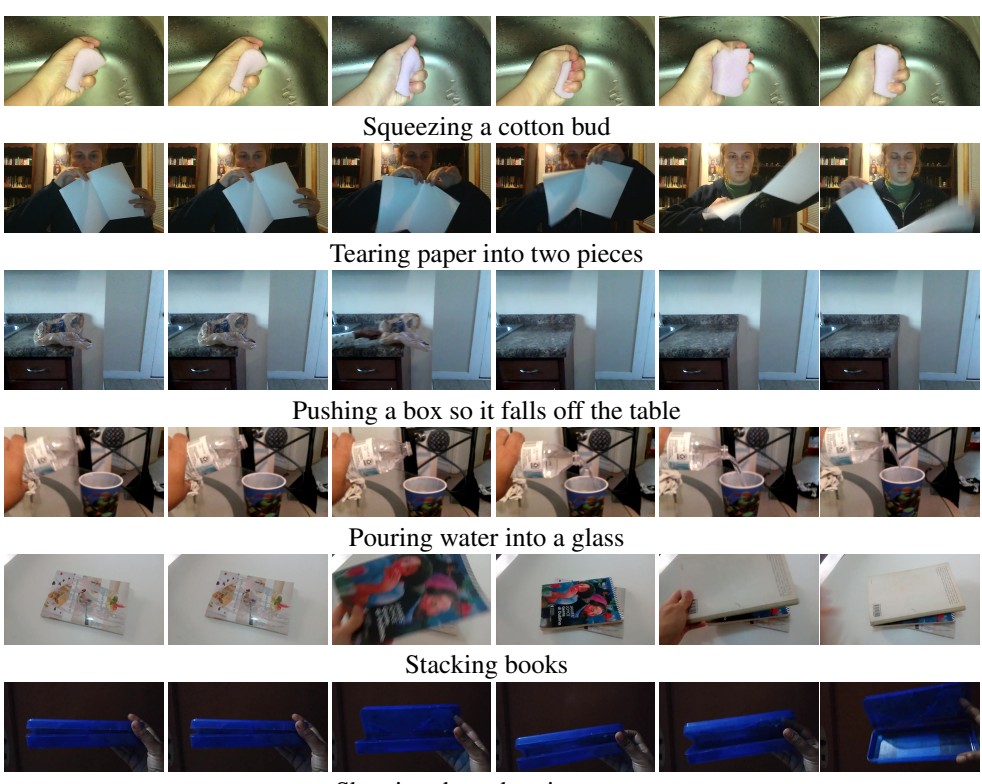

Figure 4: Examples of captioning Something-Something-V2 videos.

