# OpenReview forum: "Evaluating visual "common sense" using fine-grained classification and captioning tasks"
_ICLR.cc/2018/Workshop — Accept_

### Official Review · AnonReviewer2 · 2018-03-07
**Further discussions would have made the paper stronger**

**Rating:** 6
**Confidence:** 3

**Review:**


==========================================
Summary
==========================================

The paper proposes a new metric to measure the discriminative properties of a classification model beyond its top-k accuracy.
This metric is obtained by grouping annotated classes by different criteria, e.g. "pretending", "prepositions, "final state", etc., and reporting the mean score over groups.

In addition, an new version of the Something-Something dataset (Goyal et al,ICCV'17) is released including the group-level annotations.


----------------
Novelty
----------------

This paper proposes a new metric and a Something-Something dataset (Goyal et al,ICCV'17)

----------------
Clarity
----------------

The paper is clear and its content is easy to follow.

----------------
Significance
----------------

Beyond the proposed metric, the additionl group-related labels could be useful in other work that explores the internal properties of trained models or towards model interpretation.

----------------
Quality
----------------

Observations made through the presentated experiments are barely discussed which hinders the significance of the work. Beyond some presented results, it is hard to asses why the proposed metric indeed encondes discriminative properties of the model.

In addition, is not clear to me the purpose of the captioning problem mentioned in the paper since there is no experiment or result reported related to that problem.

==========================================
PROs:
- Clear and easy to follow.
- Potential to foster model interpretation research.


CONS:
- Findings/observations made through are barely discussed.
- No provided evidence that the proposed metric achieves its goal of measuring discriminative properties of a model
==========================================

---

### Official Review · AnonReviewer3 · 2018-03-12

**Rating:** 5
**Confidence:** 4

**Review:**

The authors present a brief study on the evaluation of visual "common sense" in video. They propose an updated dataset ('something-something' Goyal et.al. 2017), the use of 'contrastive groups' for more robust learning, and a baseline method that incorporates this.

The paper is somewhat difficult to parse -- particularly with regard to what the central thesis being proposed.

For example, it's not at all clear what the addition of object categories.
Initially, the authors state:
 "The first version of this dataset, that was introduced by (Goyal et al., 2017), is generated by asking crowd
  workers to act out template-based labels, such as “Dropping something into something” and to fill in the
  "something" placeholders with the object used to generate the video."
It's not clear what 'object used to generate the video' means, and how it is different from having object categories.

The use of contrastive groups seems like a reasonable means to clean data/evaluation to be robust to a variety of unrelated factors for the purposes considered. And the common-sense measure itself is a prior-adjusted classification score that further helps robustnes of results.

The saliency visualisations seem to indicate reasonable things; although the T-SNE plots are again hard to understand (especially Fig.2) without a proper explanation.

The captioning part is presented in too brief a form to make sense of it -- and no evaluation or references to any number of prior papers on action-recognition in videos using natural language.

I understand space is at a premium, but to cram as much as possible into a 3-page paper only makes it difficult to understand and evaluate.

Overall, I feel this paper does not quite pass muster for this venue.

---

### Official Review · AnonReviewer1 · 2018-03-13
**Paper with useful contributions but unclear in a number of important aspects**

**Rating:** 6
**Confidence:** 3

**Review:**

The paper presents the something-something v2 dataset; a dataset containing a large number of videos collected by requesting crowd workers to act out actions like "Dropping *something* into *something*". This is the second version of such dataset, in which the number of videos is doubled and introduces object categories. Apart from this dataset, the authors propose three simple baseline methods on this dataset and other relatively smaller contributions (visualization, caption prediction and a new evaluation method).

-Positive points
Extending an existing dataset is definitely a contribution to the community. The dataset is twice larger and easier for object/action separation. The inclusion of three basic baselines is very useful for researchers to test their ideas incrementally, specially when one of the baselines is better in terms of accuracy than the best current results (in dataset v1, which is not directly comparable). The paper's language is good.

- Negative points
The main negative point is the clarity of the paper. I think the authors have tried to explain too many things in a short paper, resulting in some basic ideas not being properly explained. For example:
-- Explaining the metric construction properly is *essential* for a dataset paper. However, the authors are not making it completely clear. In the computation of mA1, mA2 and the common score, is the classifier retrained so that the output only considers the classes in the contrastive group? Is the classifier left as is but only the scores within the group are taken into account? Anything else?
The metric must be perfectly unambiguous.
-- In Figure 1, what is visualized? The examples are predicted as "opening something". What are then the other annotations (Putting something into something, etc)? The ground truth? Activations related to those particular outputs?

Since the qualitative results related to the captioning and visualizations are so small compared to the size of the dataset (and therefore could be arbitrarily cherrypicked), I would prefer to remove some of the space related to them and make the rest of the paper more verbose and clear.

Summary:
I believe the paper is a step forward from the first version of the dataset. The step is relatively small, but worth being presented in a workshop. However, my rating is relatively low given the lack of clarity in essential aspects like the evaluation metric.

---

### Decision · Program_Chairs · 2018-03-20
**ICLR 2018 Workshop Acceptance Decision**

**Decision:**

Accept

**Comment:**

Congratulations, your paper was accepted to the ICLR workshop.